# Health Effects of Red Wine Consumption: A Narrative Review of an Issue That Still Deserves Debate

**DOI:** 10.3390/nu15081921

**Published:** 2023-04-16

**Authors:** Mauro Lombardo, Alessandra Feraco, Elisabetta Camajani, Massimiliano Caprio, Andrea Armani

**Affiliations:** 1Department of Human Sciences and Promotion of the Quality of Life, San Raffaele Open University, 00166 Rome, Italy; 2Laboratory of Cardiovascular Endocrinology, San Raffaele Research Institute, IRCCS San Raffaele Roma, Via di Val Cannuta, 247, 00166 Rome, Italy

**Keywords:** antioxidants, diabetes mellitus, type 2, cardiovascular diseases, body composition, inflammation, body weight

## Abstract

A strong controversy persists regarding the effect of red wine (RW) consumption and health. Guidelines for the prevention of cardiovascular diseases (CVD) and cancers discourage alcohol consumption in any form, but several studies have demonstrated that low RW intake may have positive effects on CVD risk. This review evaluated randomised controlled trials (RCTs), examining the recent literature on the correlations between acute and chronic RW consumption and health. All RCTs published in English on PubMed from 1 January 2000 to 28 February 2023 were evaluated. Ninety-one RCTs were included in this review, seven of which had a duration of more than six months. We assessed the effect of RW on: (1) antioxidant status, (2) cardiovascular function, (3) coagulation pathway and platelet function, (4) endothelial function and arterial stiffness, (5) hypertension, (6) immune function and inflammation status, (7) lipid profile and homocysteine levels, (8) body composition, type 2 diabetes and glucose metabolism, and (9) gut microbiota and the gastrointestinal tract. RW consumption mostly results in improvements in antioxidant status, thrombosis and inflammation markers, lipid profile, and gut microbiota, with conflicting results on hypertension and cardiac function. Notably, beneficial effects were observed on oxidative stress, inflammation, and nephropathy markers, with a modest decrease in CVD risk in five out of seven studies that evaluated the effect of RW consumption. These studies were conducted mainly in patients with type 2 diabetes mellitus, and had a duration between six months and two years. Additional long-term RCTs are needed to confirm these benefits, and assess the potential risks associated with RW consumption.

## 1. Introduction

Wine is an alcoholic beverage produced by the fermentation of crushed grapes. The different types of grapes and wine-making processes affect the colour and strength of the final beverage. The alcohol content varies from approximately 9 to 15% ethanol (ET) per volume [1]. Red wine (RW) contains other nutrients such as monosaccharides (e.g., glucose and fructose), variable levels of micronutrients (e.g., potassium, calcium, iron, magnesium, copper) and some B vitamins. More than 100 polyphenol compounds, including flavonoids and non-flavonoids, have been identified in RW, and to a lesser extent in white wine (WW) [2].

There is a strong ambiguity surrounding RW consumption and health [3]. Guidelines for the prevention of cardiovascular and neoplastic diseases advise against alcohol consumption, but drinking low-to-moderate amounts of wine may have some beneficial effects on cardiovascular disease risk (CVD) in certain populations [4]. Prospective cohort studies have demonstrated that any form of alcohol increases the risk of cancer [5]. In fact, the European Code Against Cancer advises limiting or eliminating alcohol consumption [6], and the International Agency for Research on Cancer has classified the consumption of alcoholic beverages as carcinogenic to humans (Group 1) in a dose-response manner. High alcohol consumption has been correlated with an increased risk of cancers of mouth, pharynx and larynx, oesophagus (squamous cell carcinoma), liver, colorectum, breast (before and after menopause), and stomach, in addition to many other diseases, such as cirrhosis, infectious diseases, CVD, diabetes, neuropsychiatric conditions, and early dementia [3,4,5].

In contrast, possible health benefits from RW intake have also been revealed, as RW contains less alcohol than spirits, and has greater antioxidant effects due to its higher polyphenol content. In fact, according to O’Keefe and colleagues, lower rates of death, T2DM, CVD, congestive heart failure, and stroke are associated with light-to-moderate habitual RW intake [1]. Even stronger results have been obtained in the framework of a healthy Mediterranean diet (MD) [2]. According to Renaud and de Lorgeril, the “French paradox” is the observation of a low prevalence of ischaemic heart disease despite a high intake of saturated fat: this phenomenon is accredited to the consumption of RW and its cardioprotective effect [7,8]. Haseeb and colleagues [9] described the composition of RW and the effects of its polyphenols on chronic CVD, and according to the authors, the polyphenols in RW could synergistically confer benefits against chronic CVD, if consumption remains within the maximum doses suggested by guidelines (one 125 mL glass for women and two glasses for men).

In consideration of the persistent contradictions, the aim of this review was to systematically evaluate the existing literature regarding the correlations between acute and chronic RW consumption and health.

## 2. Materials and Methods

We performed a PubMed search for randomised controlled trials (RCT) using the following keywords: “red” AND “wine” AND “humans”. It was decided to limit the search to only one database because PubMed’s search engine is particularly efficient, and the filters and results on other databases were duplicated. We researched papers published from 1 January 2000 to 28 February 2023. After this initial research, 6429 studies were discovered. Of these, 636 papers were excluded as they were duplicate studies. We screened 5793 records and removed studies on the following topics: cytotoxicity, functional foods, supplements (as resveratrol), probiotics, sensory aspect, dysphagia, differences in glasses or labelling, pharmacokinetics of alcohol, nasal or asthma symptoms, osmotic-stress-sensitive wine, biomarkers of wine consumption, wine extract, interaction with drugs, allergens, combination with other foods, hypothalamic–pituitary–adrenal axis, adiponectin or leptin, monocyte migration, catechin, leukocytes functions, natural wine, intelligence test, gastric emptying, diuretic effects, tailor-made beverage intervention, ET only, and hormones (e.g., aromatase). Studies in which the dose of RW was not clear, or the diet was not clearly distinguished from RW were also excluded. We thus did not consider duplicate papers, reviews, conference proceedings, abstracts, short surveys, letters, books, or book chapters. We also did not include in vitro or animal model studies. The 209 studies obtained were combined into one database. The full texts of the selected manuscripts were carefully examined. Two reviewers (M.L. and A.F.) independently judged the appropriateness of inclusion. In cases of disagreement, a third reviewer (A.A.) was invited to participate in the review procedure. Finally, 118 papers were excluded because they focused on WW, derivatives of RW, or nutraceutical or supplement use. After applying all criteria, 91 studies [10,11,12,13,14,15,16,17,18,19,20,21,22,23,24,25,26,27,28,29,30,31,32,33,34,35,36,37,38,39,40,41,42,43,44,45,46,47,48,49,50,51,52,53,54,55,56,57,58,59,60,61,62,63,64,65,66,67,68,69,70,71,72,73,74,75,76,77,78,79,80,81,82,83,84,85,86,87,88,89,90,91,92,93,94,95,96,97,98,99,100] were included in this review. Figure 1 shows the steps for applying the inclusion and exclusion criteria to determine the final group of studies considered in this review. The mean characteristics (Appendix A) and topics (Appendix A) of all papers are shown in the Appendix A.

## 3. Results

### 3.1. Antioxidant Status

The antioxidant effects of RW consumption were evaluated in 19 studies (Table 1). Moderate amounts of RW, in the context of an MD, showed beneficial effects on the oxidative status of healthy subjects due to the increased expression of antioxidant enzymes involved in the reduction in circulating ROS, such as catalase (CAT), superoxide dismutase 2 (SOD2), and glutathione peroxidase 1 (GPX1) [93]. Antioxidant effects of RW consumption have been described for various conditions, including type 2 diabetes mellitus (T2DM) and acute CVD, as well as in elderly people [14,17,29,34,50]. This protective effect seems to be related to its high polyphenolic content. The plasma concentration of polyphenols was particularly higher in individuals consuming RW, compared with those consuming WW, resulting in the higher inhibition rate of oxidative stress. Of note, the described antioxidant effect may be dependent on a synergism between the different polyphenols contained in RW [32]. Accordingly, RW may have more powerful antioxidant effects than gin [59] and WW [56], for example, which have a lower content of polyphenols. The antioxidant effect of RW is also mediated by other bioactive compounds. It has been shown that moderate doses of RW increase blood concentrations of homovanillic acid, a typical biomarker of dopamine turnover, in healthy volunteers. Homovanillic acid is also a phenolic metabolite, so the increased blood concentrations of this biomarker could result from the metabolism of RW phenolic compounds [85]. RW’s antioxidants also suppress the post-prandial activation of NF-κB, an oxidative-stress-related transcription factor involved in the regulation of inflammatory responses [17,70]. Similarly, oxidized guanine species and protein carbonyl levels were significantly decreased in the RW group in CAD patients [100]. Importantly, it is well established that the beneficial effects observed after RW consumption, in terms of the increased activity of antioxidant enzymes, are not due to the alcohol content in wine, but to the polyphenolic composition. De-alcoholised red wine (DRW) could therefore be an excellent source of antioxidants with protective properties in conditions linked to oxidative stress [68]. On the other hand, plasma antioxidant capacity, measured through the ferric reducing antioxidant power (FRAP) in healthy subjects, was similarly increased by DRW and RW deprived of polyphenols. The authors hypothesised that increased FRAP after polyphenol-stripped RW consumption could be primarily explained by the increase in plasma urate, which is also a strong antioxidant with free radical scavenging and metal chelating capabilities [51]. Conversely, Blackhurst et al. demonstrated that RW consumption increased plasma antioxidant catechin levels, but did not affect the postprandial peroxidation status of chylomicrons after a high-fat meal. Moreover, RW did not increase the oxygen radical absorbance capacity (ORAC) in fasting subjects after a meal [39]. Interestingly, one glass of RW displayed lower amounts of bioavailable flavonols than one glass of tea or 15 g of red onion (RO), although urinary excretion of quercetin after wine consumption did not differ from that measured after onion consumption and was higher than that after tea, suggesting that RW has similar antioxidant activity and could therefore prevent LDL oxidation [15]. In contrast, Chiu et al. demonstrated that RO has greater antioxidant protection on plasma LDL [88]. It is notable that the alcohol content of 375 mL RW/day increased oxidative stress in a 4-week RCT [90]. Similarly, Addolorato and colleagues demonstrated that ET could increase lipid peroxidation parameters and reduce antioxidant capacity, although these effects were attenuated when ET was consumed in beer or RW [45].

### 3.2. Cardiovascular Function

The effects of RW consumption on cardiovascular function were evaluated in seven studies (Table 2). Acute RW consumption, in comparison with gin, had a greater down-regulatory effect on genes related to atherosclerosis progression in men with high CVD risk, probably due to the higher phenolic content [96]. In patients with stable angina pectoris, the acute intake of low doses of RW did not significantly affect ischemic pre-conditioning (IPC) during exercise stress testing [57]. Acute RW and DRW consumption did not improve coronary epicardial diameters or flow rate, even though both beverages reduced vasoconstrictive peptide endothelin-1 levels [64]. Moderate consumption of RW over 1–2 weeks did not reduce CVD risks by altering either the coronary microcirculation or haemorrheology [62]. In a two-year randomised controlled trial, moderate wine intake did not affect total carotid plaque volume, although the subjects with the greatest baseline plaque severity displayed a small regression in plaque burden [94]. In young healthy individuals, low blood concentrations of ET following RW intake had an acute depressant effect on left ventricular (LV) performance, but an increase in some indices of right ventricular function [65], suggesting that low ET doses may impair LV function. A lifestyle modification intervention (including RW consumption) did not affect the blood flow velocity of the internal carotid or middle cerebral artery in patients with carotid atherosclerosis [76]. Most of the enrolled patients were receiving statin therapy, however, which could have hidden the beneficial effects on blood flow velocity and thus affected the study results. Overall, low consumption of RW did not significantly improve CVD parameters even in long-term studies, suggesting that other factors modulated by RW account for the CVD risk. None of the RCTs included in this review assessed the effects of RW on atrial fibrillation and other cardiac arrhythmias.

### 3.3. Coagulation Pathway and Platelet Function

The effect of RW consumption on coagulation and platelet function was evaluated in eleven studies (Table 3). These studies were conducted for up to three weeks, although one study ran for three months [20]. The intake of RW during a meal significantly decreased thrombotic activation, in terms of the plasma levels of prothrombin fragments 1 + 2 and activated factor VII [14], and fibrinogen concentrations [41], indicating inhibitory effects on the coagulation system. A sustained viscosity-reducing effect of plasma was observed after three weeks of RW consumption. A reduced viscosity was retained after three weeks of abstention [41]. RW intake led to a significant reduction in the aggregation ability of platelets against platelet activating factor (PAF), regardless of the ET percentage content [89]. RW decreased [17] or did not affect [87] concentrations of plasminogen activator inhibitor-1 (PAI-1), a risk factor for thrombosis and atherosclerosis, more negatively than ET itself [87]. RW and beer could reduce von Willebrand factor (vWF) levels. The vWF factor promotes the adhesion of platelets to damaged blood vessel walls and then acts as a bridge between one platelet and another, promoting clot formation. RW consumption did not increase the PAI-1/tPA ratio, which is associated with increased CV risk, an effect observed after the intake of other alcoholic beverages [58]. Thus, RW, but not ET alone, reduced the specific activity of platelet-activating factor and RW had a weak but significant inhibition effect on platelet activation. This RW effect is greater than that of WW, probably due to the higher polyphenol content [21]. Three studies did not demonstrate that RW had a favourable effect on coagulation and platelet function. The consumption of RW or WW with dinner did not affect the platelet count, platelet function, or viscoelastic properties of the blood taken the next morning [23]. In a study comparing a MD with a high-fat diet, the moderate consumption of RW resulted in a significant increase in platelet aggregation and secretion, but did not significantly modify the bleeding time or plasma vWF concentrations [20]. Similarly, there were no differences in fibrinogen and D-dimer levels after two weeks of drinking RW [62]. In contrast, Banach et al. observed a significant increase in PAI-1 concentration in the RW-drinking group [72]. Overall, RW intake shows the ability to reduce platelet aggregation and the process of coagulation.

### 3.4. Endothelial Function and Arterial Stiffness

The effect of RW consumption on endothelial function was evaluated in 15 studies (Table 4). The consumption of RW may have positive effects on endothelial function according to most of the studies considered. These effects, in term of increased flow-mediated dilatation (FMD), have been clearly demonstrated in patients with hypercholesterolemic [27] and CAD [25]. Polyphenols were shown to play an important role in inducing RW-mediated acute vasodilation and a reduction in von Willebrand factor levels in healthy subjects [40,58]. These data suggest that polyphenols play a role in mediating RW effects on endothelial function [40]. Notably, in healthy subjects, a daily consumption of 100 mL of RW for three weeks led to increased levels of circulating endothelial progenitor cells (EPC), probably mediated by the increased bioavailability of plasma nitric oxide, with potential cardiovascular protective effects. In vitro studies showed that resveratrol could recapitulate the effects of RW on EPC function [61]. When flow-mediated vasodilation was examined, a single dose of DRW increased endothelium-dependent vasodilation in response to hyperaemia, whereas ingestion of RW induced vasodilation without affecting the percentage increase in artery diameter [10]. ET consumption dilates the brachial artery and increases muscle sympathetic nerve activity, heart rate, and cardiac output with dose-dependent effects. These acute effects are not modified by RW polyphenols [54], suggesting that the content of polyphenols in RW, rather than ethanol, exerts cardiovascular protective actions [27]. Accordingly, RW’s antioxidants can counteract the acute effects of smoking on the endothelium [48].

Other studies have shown that RW has no effect or an unfavourable effect on endothelial function. Intake of a moderate amount of RW did not increase endothelial function [29]. In postmenopausal dyslipidaemic women, the intake of DRW and RW for six weeks was not associated with significant changes in vascular function or arterial stiffness, in comparison with women consuming water [43]. Similarly, the regular daily consumption of RW for a four-week period did not alter endothelial function [36]. Of note, Banach and colleagues demonstrated that a high acute consumption of RW in a previously abstinent population can lead to a significant increase in plasma concentrations of a potent vasoconstrictor (endothelin-1) [72]. Similarly, an intake of 375 mL RW/day for four weeks increased vasoconstrictor eicosanoids [71,90]. Although rapid alcohol consumption causes vasodilation at the level of the distributing artery as well as at the arteriolar level, a reduction in FMD has been observed in subjects consuming RW or an alcoholic beverage with a low content of polyphenols, suggesting that the polyphenols contained in RW may not be able to preserve the endothelial function [49].

### 3.5. Hypertension

The effect of RW consumption on blood pressure (BP) was evaluated in 13 studies (Table 5). Six studies lasted up to four weeks [36,54,66,69,78,86,90]. Only one study evaluated the effect over a longer time span of six months [79]. The ingestion of 250 mL RW at lunch resulted in a reduction in postprandial BP in subjects with hypertension and central obesity [18]. Consumption of either RW or DRW resulted in decreased systolic blood pressure (SBP) in patients with coronary artery disease [31]. These data were confirmed in another study that evaluated the effects of RW in combination with smoking on haemodynamic parameters. Both RW and DRW were able to prevent the increase in peripheral SBP induced by smoking. Notably, both RW and DRW were able to decrease postprandial wave reflexes, with RW having a more pronounced effect, suggesting that the alcohol present in RW contributes to reducing the augmentation index [44].

Interestingly, Chiva-Blanch et al. [66] demonstrated that DRW consumption for four weeks, compared to RW or gin, led to a reduction in SBP and diastolic blood pressure (DBP), with a parallel increase in plasma NO concentration, which may mediate the observed effects on BP.

Other studies have found no effect from moderate RW consumption on BP. In patients with T2DM who were alcohol abstainers, consuming RW for six months had no effect on their mean 24-h BP. A more pronounced BP-lowering effect was noted among fast ET metabolisers, however [79]. In subjects consuming different beverages (i.e., RW, ethanol, water) no intervention was able to affect blood pressure [54]. RW containing 24–31 g of alcohol per day, administered for four weeks, increased 24-h BP and heart rate (HR) but reduced waking BP in well-controlled T2DM subjects [86].

RW was described as having adverse effects on BP by Barden and colleagues, who showed that a dose of 375 mL RW/day for four weeks increased BP, plasma levels of CYP450 vasoconstrictor eicosanoids and oxidative stress [90]. It has been suggested that increased plasma levels in the vasoconstrictor 20-HETE promote BP elevation and potentially contribute to the BP elevation associated with a binge drinking pattern [71]. Similarly, regular consumption of 200–300 mL RW/day in healthy premenopausal women raised HR and 24-h SBP and DBP. In the DRW group, BP values were similarly elevated [82], and accordingly, Zilkens et al. also demonstrated that polyphenols do not play a significant role in mitigating the effects of alcohol in increasing BP in men [36]. Other authors have shown that postprandial RW ingestion has a moderate effect on increasing DBP, but not SBP [78].

### 3.6. Immune Function and Inflammatory Status

The effects of RW consumption on the immune system and inflammatory status were evaluated by 14 studies (Table 6). Most were conducted for a period of up to four weeks, and only one study was conducted for 12 weeks [92]. The serum levels of pro-inflammatory cytokines (IL-6 and TNF-a) and serum levels of acute-phase proteins (hs-CRP and fibrinogen) did not show a significant change after the acute consumption of various alcoholic beverages, including RW [58]. Watzl et al. also reported that postprandial consumption [19] and moderate daily consumption for two weeks [22] of RW or DRW in healthy men had no negative effects on human immune cell function. Similarly, RW did not affect the plasma levels of lipid mediators of inflammation resolution (SPMs) in patients with T2DM [92]. No significant changes were demonstrated in the plasma levels of hs-CRP, IL-6, IL-18, VCAM-1, CASP-1, MMP-9, TIMP-1, APO B, cystatin C, or ICAM-1 [95].

Avellone et al. [28] showed that LDL/HDL, fibrinogen, factor VII, plasma C-reactive protein, and antibodies to oxidised LDL were markedly reduced, while HDL-C, Apo A1, TGFbeta1, t-PA, PAI, and total plasma antioxidant capacity were significantly elevated, indicating the beneficial effects of RW on levels of CV risk biomarkers. Conversely, the study by Banach et al. [72] demonstrated a negative effect on the fibrinolytic system and endothelial function (increased tPA:Ag, PAI:Ag and E-1) of consuming different types of alcoholic beverages, including RW, on haemostatic factors. Several studies have shown that the intake of RW led to a decrease in inflammatory or immune biomarkers in the serum. Moderate alcohol consumption, such as RW, marginally reduced fibrinogen levels in healthy subjects [33]. RW was more effective than WW in promoting a reduction in serum inflammatory biomarkers, CAM expression on monocyte surface membranes, and monocyte adhesion to endothelial cells [51]. Another study detected favourable effects of RW only in participants with high cytokine levels, in particular for cytokines that promote initial inflammation such as TNF-α, IL-6, and IFN-γ [77]. Furthermore, lower secretion of ΤNFα was observed after 8 weeks of intake in the RW group versus the ET group [97]. A Cava rosé wine, with a medium-level polyphenol content, has been shown to reduce the inflammatory markers of atherosclerosis (adhesion molecules, cytokines, and the CD40/CD40L system) to a greater extent than other alcoholic beverages without polyphenols [55]. Consumption of 375 mL of RW/day for four weeks also increased SPMs, in particular, resulting in higher levels of 18 -HEPE, RvD1, and 17R-RvD1, capable of attenuating excessive inflammation [90]. In contrast to the other studies, Williams and colleagues demonstrated that, in men with CVD, the consumption of moderate amounts of RW acutely increased plasma IL-6 levels, probably in response to alcohol-induced oxidative stress in the liver [26].

### 3.7. Lipid Profile and Homocysteine Levels

The effects of RW consumption on the lipid profile were evaluated by 23 studies (Table 7). The moderate consumption of RW could have a modest beneficial effect on lipoproteins and cellular cholesterol efflux compared with an alcohol solution [12]. Accordingly, alcohol consumption could promote a reverse cholesterol transport pathway (RCTP), the process by which cholesterol is directed from the peripheral tissues to the liver via HDL for subsequent excretion in the bile, regardless of alcoholic beverage type (RW, beer, or spirits) [16]. Similarly, Chiva-Blanch et al. reported that RW may reduce plasma concentrations of lipoprotein (a), which is responsible for transporting cholesterol in the blood and is considered a CVD risk factor that does not respond to standard therapy for low density lipoprotein (LDL) reduction [67]. Two Sicilian RWs were shown to have positive effects on several cardiovascular risk factors, including HDL and Apo A1, in a total of 48 subjects of both sexes [28]. A study by Tsang and colleagues also revealed that oxidised LDL levels were reduced, while HDL cholesterol concentrations increased modestly after RW consumption in healthy volunteers [34]. Another study reported a 34% reduction in LDL in the RW group, compared with controls [47]. A randomised cross-over trial comparing the effects of a moderate intake of RW with alcoholic beverages without polyphenols on plasma antioxidant vitamins, lipid profile, and the oxidability of LDL particles, demonstrated that RW would provide additional lipid benefits due to its antioxidant effects by decreasing plasma levels of MDA, SOD, and oxidised LDL [59]. Notably, RW daily consumption associated with lifestyle changes, including moderate physical exercise for 20 weeks, improved the LDL/HDL ratio in patients with carotid atherosclerosis [74]. A study investigating the potential contribution of ET components in inducing beneficial effects on the lipid profile level demonstrated that RW consumption for four weeks induced an improvement in HDL and fibrinogen plasma levels compared with control groups drinking water, with or without red grape extract [30]. In a recent study by Briansó-Llort and colleagues, RW rich in resveratrol had positive effects on total cholesterol, in healthy volunteers [98].

Other studies did not demonstrate beneficial effects of RW consumption on lipid metabolism [27,100]. The ingestion of DRW, but not RW, decreased circulating F2-isoprostanes, a prostaglandin-like compound formed from the free-radical-mediated oxidation of arachidonic acid [13]. It was supposed that phenolic compounds would have a beneficial effect on lipid peroxidation if consumed separately from alcohol. Similarly, consumption of DRW containing 880 mg total polyphenols in dyslipidaemic postmenopausal women did not change triglyceride levels or postprandial chylomicrons. The same study demonstrated a 35% increase in postprandial TG levels following RW consumption, compared to the control group [24]. An intake of 300 mL RW did not reduce LDL oxidation in healthy subjects [35] and did not affect lipid peroxidation in postprandial chylomicrons [11,39]. RW consumption did not lead to any increase in HDL and the lipid profile was unchanged in RW vs. WW [91]. No change in serum HDL cholesterol was observed in diabetic patients with consumption of RW (24–31 g alcohol/day) compared to DRW or water [86]. Similarly, in subjects with T2DM, daily intake of 150 mL of muscadine grape wine (MW) or de-alcoholised MW had no significant effect on the lipid profile compared to the other groups [37]. LDL-cholesterol levels were reduced in subjects without steatosis at baseline, but there were no changes in HDL or triglycerides levels with moderate RW consumption for three months. In contrast, hepatic triglyceride content increased [63]. An intake of RO extract, which is very rich in polyphenols, showed greater cholesterol-lowering efficacy than RW [88].

RW intake was not observed to elevate plasma homocysteine levels in T2DM subjects [86]. Conversely, 24 g/day alcohol given as RW to healthy men for two weeks significantly increased homocysteine levels [56].

### 3.8. Body Composition, Type 2 Diabetes, and Glucose Metabolism

The effect of RW consumption on body weight, body composition, and adipocytokines levels was evaluated by six studies (Table 8). When part of an energy-restricted regimen, moderate RW consumption, with or without alcohol, improved glycaemic control in diabetic patients, with a parallel improvement in several metabolic parameters related to antioxidant status [37]. Accordingly, moderate wine consumption did not promote weight gain or abdominal adiposity in controlled diabetic patients following the Mediterranean diet [84], nor did fat accumulate in subcutaneous and abdominal fat depots in healthy subjects with a waist circumference above 94 cm [38]. A study that focused on two types of RW with different resveratrol content showed that the consumption of neither RW changed BMI [98]. Interestingly, healthy subjects consuming RW for 14 days displayed increased plasma levels of leptin, the main adipokine with the primary function of regulating energy balance, while no significant increase in plasma levels of adiponectin and other adipocytokines was observed [95]. Djurovic et al. revealed that RW could induce an increase in circulating leptin levels in females, but not in males [46].

Thirteen studies evaluated the effects of RW consumption on glucose metabolism (Table 9). A significant decrease in insulin and an increase in the fasting glucose-to-insulin ratio were demonstrated among T2DM subjects given DRW [37]. In another study, the RW group exhibited lower insulin levels and HOMA scores compared with the control group, while other metabolic parameters were similar [42]. Increased oxidative stress may play a role in the progression of diabetic nephropathy, a microvascular complication of diabetes. It has been demonstrated that RW consumption reduces urinary protein levels, as well as urinary 8-OHdG and urinary L-FABP, in patients with diabetic nephropathy, thus providing protective effects against the clinical progression of chronic kidney disease [60]. These results suggest that RW, but not WW, has protective effects in diabetic patients, probably due to its ability to reduce oxidative stress. In another study, fasting blood glucose remained unchanged, while basal insulin and HOMA-IR values decreased significantly in the groups in which RW and DRW (30% and 22%, respectively) were consumed compared to gin [67]. A moderate intake of RW in adults with obesity and metabolic syndrome (MetS) resulted in a reduction in MetS risk markers and improved gut microbiota composition [81]. Moderate RW consumption significantly reduced fasting glucose levels, especially in subjects with higher basal A1C levels, but not postprandial levels: the reduction in A1C levels was not statistically significant [53]. On the other hand, RW consumption during a meal induced an increase in plasma glucose and insulin similar to that observed after a meal without wine in diabetic patients. Nevertheless, wine ingestion with the meal counterbalanced the decrease in plasma levels of total radical antioxidant parameters (TRAP) induced by food in the postprandial phase, with beneficial effects on LDL oxidation and thrombotic activation in these patients [14].

The insulin sensitivity index (ISI), assessed with a hyperinsulinaemic euglycemic clamp, was not affected by moderate RW consumption, in healthy subjects with visceral fat accumulation [38]. Interestingly, although both RW and WW tended to improve the glucose metabolism after two years in diabetic patients, only WW significantly reduced plasma glucose levels and HOMA-IR scores, compared to the control group [79]. Notably, moderate RW consumption did not alter the plasma mediators of inflammation in patients with T2DM [92]. No significant differences in fasting plasma glucose levels were revealed in a 14-day pilot study that compared two types of RW with different resveratrol contents [98]. Thus, no fasting glycaemic changes were detected in CAD [100] and hypercholesterolaemic [27] patients after ingesting RW. Taken together, these results suggest that RW consumption in association with a meal may improve glycaemic control in diabetic patients, without affecting the body weight of patients with T2DM.

### 3.9. Gut Microbiota and Gastrointestinal Tract

Seven studies evaluated the effects of RW consumption on the gut microbiota and gastrointestinal tract (Table 10). Moderate consumption of RW (16 g ethanol/day) over three months promoted an increase in hepatic triglyceride content, without developing steatosis, in healthy subjects. Interestingly, the authors also observed a significant reduction in LDL cholesterol levels [63]. A clinical study focused on the effects of different alcoholic beverages on the postprandial functionality of the digestive system in terms of gastric emptying kinetics and orocaecal transit, and demonstrated that RW had an inhibitory effect on the gastric emptying of solid food, as well as on meal-induced gallbladder emptying, compared with beer and whisky, in healthy volunteers. No effect was observed on the orocaecal transit time of food [75]. These results suggest that RW, obtained by fermentation only, affects the motor functions of the digestive tract and of the gallbladder differently from other alcoholic beverages, depending on the ET concentration.

Three studies evaluated the effects of RW consumption for four weeks on the gut microbiota. First, RW polyphenols inhibit non-beneficial bacteria from the human microbiota and promote the growth of probiotic bacteria such as bifidobacteria in healthy male volunteers. RW particularly modulated the growth of select beneficial gut bacteria species, with a parallel improvement in blood pressure, lipid profile, and inflammatory status, linked to changes in the bifidobacteria number [69]. Another study demonstrated that RW consumption induced a significant increase in amounts of Bifidobacterium and Prevotella with a negative correlation with lipopolysaccharide (LPS) plasma concentration [73]. Similar results in terms of increased faecal microbial diversity, have been obtained both in healthy individuals [81,99] and in patients with metabolic alterations associated with obesity [83]. Plasma trimethylamine N-oxide (TMAO) did not differ between people receiving the RW intervention and ET abstention [99].

Taken together these studies suggest that the inclusion of RW polyphenols in the diet could provide prebiotic benefits for the gut microbiota through stimulating the growth of beneficial species, including Enterococcus, Prevotella, Bacteroides, and Bifidobacterium, with a positive effect on health.

## 4. Discussion

The Global Burden of Disease consortium estimated that 1.34 billion people consumed harmful amounts of alcohol in 2020 (1.03 billion males and 0.312 billion females) [101]. It is also relevant to note that in the first two years of the COVID-19 pandemic, eating and lifestyle habits changed globally [102]. A recent study by Cicero and colleagues showed a significant increase in daily RW consumption in Northern Italy during the lockdown period (February–April 2020). As RW consumption had been increasing even before the COVID-19 pandemic, it is worth emphasising that long-term ET consumption has been associated with acute and chronic negative effects on health [103]. The deleterious effects of heavy alcohol consumption have been established by numerous studies, and led to an increased risk of driving accidents [104], infections [105], violence [106], foetal alcohol disorders [107], and a high risk of acute CVD events [108]. Most common diseases with a high morbidity and mortality, such as CVD [109], T2DM [110,111], cancer [112], and digestive diseases [113], are related to the total amount of ET ingested, the pattern of consumption, and the type of alcoholic beverage consumed. For most diseases, dose–response curves grow from zero consumption upwards, with many exponential curves [114]. It is therefore evident that even the occasional consumption of beverages with a high ET content (e.g., spirits) and/or chronic abuse of any alcoholic substance is deleterious, and that there is no beneficial reason to start consuming any kind of alcoholic beverage [115,116].

It is crucial to establish the effects of chronic consumption of RW, which has a lower ET content than spirits and a higher flavonoid content than WW, beer, and spirits. Two main questions remain unanswered: (a) Does RW have the same deleterious health effects that have been widely demonstrated for other alcoholic beverages? (b) Do the benefits of resveratrol and flavonoids in RW outweigh the toxic effects of ET? A prospective cohort study by Jani and colleagues showed that consumers of spirits and beer, compared with those who consume RW, had a higher absolute and relative risk of mortality, of experiencing a major adverse CVD event, of liver cirrhosis, and of accidents/self-harm. The relative risk was even higher in subjects who drank alcohol without eating [117].

The analysis of the 91 papers included in this review (Appendix A) show the positive effects of RW consumption on antioxidant status, thrombosis, immune function and inflammation, lipid profile, and gut microbiota, whereas neutral effects were observed on body weight and glucose metabolism (Table 1, Table 2, Table 3, Table 4, Table 5, Table 6, Table 7, Table 8, Table 9 and Table 10). Beneficial effects from RW, in term of oxidative stress, inflammation, and nephropathy markers, with a modest reduction in CVD risk, were observed in five of seven studies analysed, which were performed mainly with T2DM subjects who consumed RW for at least six months [42,60,79,80,94] (Appendix A). These seven studies did not detect changes in blood pressure, weight gain, or blood glucose levels, however. There are no RCT that have evaluated the effect of RW on the risk of arrhythmia. Some cohort studies have suggested that RW, when consumed in moderation, may have a protective effect against atrial fibrillation [118], while others have found no significant effects or even potential harm [119,120,121]. A study published in the Journal of the American College of Cardiology [119] demonstrated a J-shaped relationship between total alcohol consumption and risk of AF in a population of middle-aged and older adults. Drinking RW or WW appears to be potentially safer than beer or spirits [119,120]. The dose of RW also seems to be relevant in influencing the risk of arrhythmia [121]. Consumption of two or more RW drinks per day was associated with a small but statistically significant increased AF risk [122].

A review published in 2017 revealed that light to moderate RW consumption is beneficial for the heart [7]. In contrast, a study published two years ago concluded that even low amounts of ET increase the risk of cancer and early death [123]. It is also possible that observational studies overemphasise the positive effects of alcohol on CVD outcomes. Although moderate RW intake is presumably linked to a low number of CVD events, there are many confounding elements, especially genetic and socioeconomic associations with RW consumption, that probably explain most of the association between wine and a reduction in CVD events [124]. It is also important to consider other lifestyle variables that might confound the data. For example, it was found that RW drinkers tended to buy healthier food than beer or spirit drinkers; this tendency of RW drinkers to eat according to nutritional guidelines or have healthier attitudes could explain some of the positive health effects in these subjects [115,125,126]. On the other hand, the specific contribution of RW was emphasised in a longitudinal study from the UK Biobank that showed a lower visceral fat mass (β = −0.023, *p* < 0.001) in subjects who drank more RW, associated with reduced inflammation and increased HDL. In fact, RW may help to protect against adipogenesis due to its anti-inflammatory/eulipidaemic effects [127].

Most of the studies included in our review suggest that the health benefit of RW is due to polyphenol intake. RW naturally has a high concentration and wide variety of polyphenolic compounds. The considerable presence of polyphenols has led some foods to be defined ‘superfoods’ in order to emphasise their remarkable health benefits. Dietary polyphenols could have benefits for the prevention of MetS, different forms of cancers, and neurodegenerative diseases [128]. Foods rich in flavonoids, such as berries and RW, have thus been associated with a lower risk of mortality [129]. Flavonoids, due to their antioxidant, anti-atherogenic (inhibition of LDL oxidation), and antithrombotic (reduction in platelet aggregation) effects, may have a significant role in preventing atherosclerosis and thrombosis [130]. Patients imbibing drinks with a higher amount of polyphenols, adjusted for potentially confounding factors, had a significantly lower risk of hypertension [131]. Interestingly, the anti-inflammatory effects of RW might be increased by polyphenols combined with ET. An analysis of the cardiovascular contribution of polyphenols and alcohol in RW showed that phenolic compounds could reduce the levels of serum factors, such as intercellular adhesion molecule-1 and E-selectin, which mediate the process of leukocyte adhesion. In contrast, both ET and RW polyphenols could contribute to reducing soluble inflammatory mediators such as CD 40 ligand, IL-16, and MCP-1 [132]. A possible role has been suggested for ET itself as a potential cardioprotective agent. Many epidemiological studies have confirmed a J-shaped curve for ET in a similar way as for the consumption of foods or drinks rich in polyphenols [133,134]. However, a systematic review by Yoon and colleagues, suggested that the presence of comorbidities or age may affect the protective actions of alcohol consumption on CVD [135]. Several papers in our review [30,45,54,65,71,82,90] evaluated the effects of ET in RW. ET increased oxidative stress [90] and reduced antioxidant capacity [45], with detrimental effects on cardiac function through impaired LV function [65], dilation of the brachial artery [54], and increased muscle sympathetic nerve activity [54]. ET increases heart rate [54,82] and cardiac output with dose-dependent effects [54], and may contribute to BP elevation [71,82,90] through a stimulation of CYP450 vasoconstrictor eicosanoids [90]. Conversely, one study showed that ET in RW could have a positive effect on CVD risk reduction through an increase in HDL levels [30]. The potential toxicity and risk of addiction induced by ET might mean that the consumption of polyphenols through foods is favoured over RW. High concentrations of polyphenols are found in spices and herbs, cocoa, berries, seeds such as flaxseed, nuts such as hazelnut, olives, and some vegetables such as artichoke [136]. Due to its quercetin 3-*O*-glucoside, epicatechin, and epigallocatechin gallate content, and other compounds, green tea has higher antioxidant activity than RW [137]. Cocoa shows higher levels of phenolic compounds (611 mg of gallic acid equivalents, GAE) than black tea, green tea, and RW [138].

On the other hand, the advantages of RW consumption include the greater complexity of the polyphenol composition, due to enzymatic and chemical reactions during fermentation, which distinguish RW from other sources of polyphenols, including grape juice [139]. The RW content of melatonin, with antioxidant, anticarcinogenic and cardioprotective properties, should be also considered [140,141]. Finally, RW has been shown to stimulate the growth of beneficial bacteria present in the gut microbiota, such as Bifidobacterium and Prevotella, with a subsequent reduction in plasma levels of LPS, whose increase is associated with a high risk of insulin resistance and CVD [69,73]. RW demonstrates the ability to affect the gut microbiota composition by increasing the amount of bacteria producing butyrate and reducing the levels of bacterial organisms producing LPS, with protective effects against risk factors for metabolic syndrome [81]. TMAO has been identified as a potential CVD risk factor linked to diet, the gut microbiota, and cardiovascular health. Studies have shown that plant-based diets, such as the MD, vegetarian, and vegan diets, can effectively improve TMAO levels, while animal-based diets appear to have the opposite effect [142,143]. The only recent RCT included in the review [99] that evaluated TMAO concluded that TMAO concentrations were unrelated to RW intake. The addition of RW to a plant-based diet would offer no further benefit in reducing blood and urine values of TMAO, as suggested in a previous paper [144].

Finally, we acknowledge that this review has some limitations. The research was restricted to one database, and only English-language RCTs were included; nevertheless, a large number of studies were identified.

## 5. Conclusions

Acute and short-term RW consumption seems to exert positive effects on antioxidant status, the lipid profile, thrombosis and inflammation markers, and the gut microbiota. Importantly, a longer duration of treatment with RW has been shown to protect renal and cardiac function parameters in T2DM patients, suggesting that a moderate intake of RW may serve as a dietary supplement in diabetic patients. On the other hand, blood pressure values, homocysteine levels, and gastrointestinal function seem to be impaired by short-term RW intake. The studies selected in this review confirm that the RW intake levels may influence the protection against cardiovascular and metabolic diseases. The deleterious effects observed for ET consumption limit the dose of RW that can be consumed, and additional studies might clarify whether different dosages of RW preserve metabolic and cardiovascular functions of at-risk subjects, such as diabetic patients. This review also suggests that the intake of DRW may be a promising strategy to prevent cardiovascular and metabolic dysfunction. The removal of ET may exclude the detrimental effects of RW consumption, due to the potential risks of addiction and chronic illness, and maintain the content in flavonoids, preserving their protective and healthy actions.

## Figures and Tables

**Figure 1 nutrients-15-01921-f001:**
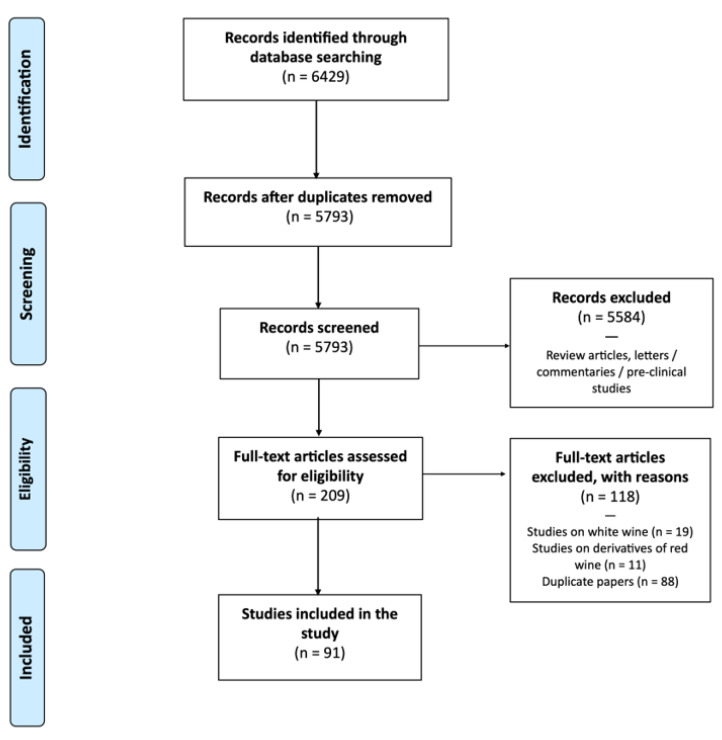
Flow chart of studies included.

**Table 1 nutrients-15-01921-t001:** Overview of studies that evaluated the antioxidant effects of red wine consumption.

Ref.	Year	Author	Antioxidant Effects	Indicators of Total Antioxidant Capacity	Endogenous Oxidation Biomarkers	Nucleic Acid Oxidation	Lipid Peroxidation	Other Antioxidant Effects	Type of Study	Type of Wine	Subjects	Patients	Age Range	Total Number of Participants	Wine Consumption Duration (Days)	Control	Dosage (mL/Day)	Funding
[14]	2001	Ceriello A	↑ ç	↑ (TRAP) ç			↓ ox-LDL ç		RCT	RW	M + F	T2DM	50–60	20	7	W	300	Not declared
[15]	2001	de Vries JH	=	= &					RCT	RW	M	H	20–35	12	4	50 g fried onions or 375 mL of black tea	750	Netherlands Heart Foundation
[17]	2002	Mansvelt EP		↑ (TAS)					RCT	RW	M + F	H	25–56	13	28	WW	ET: 23 g for F, 32 g for M	Not declared
[29]	2005	Guarda E	↑	↑ (FRAP), ↑ (TAS)		↓ 8-OH-dGuo			RCT	RW	M + F	CVD patients	55–62	20	60	W	250	Not declared
[32]	2005	Pignatelli P	↑				↓ PGF2	↓ Isoprostanes £, ↓ PKC-mediated NADPH oxidase activation	RCT	RW	M + F	H	35–50	20	15	WW, W	300	Not declared
[34]	2005	Tsang C	↑				↓ (TBARS), = ox-LDL		RCT	RW	M + F	H	23–50	20	14	W	375	Not declared
[39]	2006	Blackhurst DM	=	= (ORAC, LOOH)			= (TBARS)	↑ Catechins	RCT	RW	M + F	H	25–45	15	acute	W	M: 230, F: 160	Wine Industry of South Africa
[45]	2007	Addolorato G	↓		↑ MDA, ↓GSH		↑	↓ Vitamin E	RCT	RW	M + F	H	20–30	30	30	beer, spirit	400; 11% ET	Association for Research in Medicine’ Foundation Bologna-Rome(Italy).
[50]	2007	Micallef M	↑	↑ (TAS)	↓ MDA, ↓ GSH		↓		RCT CO	RW	M + F	H	18–30	20	14	W	400	Not declared
[51]	2007	Modun D	↑	↑ (FRAP) $	↑ plasma urate $				RCT CO	RW	M	H	25–40	36	acute	DRW, PSRW, W, ET	195–280	Ministry of Science, Education and Sports of the Republic of Croatia
[59]	2009	Estruch R	↑		↓ MDA, ↓SOD		↓ ox-LDL	↑ Vitamin E	RCT CO	RW	M	H	30–50	40	28	gin	30 g/ET	Institutional
[60]	2009	Nakamura T	↑ °			↓ 8-OHdG		↓ L-FABP	RCT	RW, WW	M + F	T2DM with nephropathy	45–65	20	6 months	W	118	Not declared
[68]	2012	Noguer MA	↑ §	↑ (FRAP) §	=GSH/ GSSG				RCT CO	DRW	M + F	H	25–40	8	14	LPD	300 (+low phenolic diet)	Institutional
[70]	2012	Schrieks IC	↑ = (TEAC)	↑ (TAS)			↑ 8-iso-PGF2α #	↓ NF-κB activation 8-iso-PGF2α.	RCT CO	RW	M	Overweight	35–68	19	30	DRW	450 (41.4 g ET)	Dutch Foundation for Alcohol Research
[85]	2016	Marhuenda J	↑			↓ 8-OH-dGuo ↓ 8-OH-GUa ↓ 8-NO2-Guo		↑ Homovanillic acid	RCT CO	RW	F	H	18–27	9/9	37	RW (3 types)	200	Institutional
[88]	2016	Chiu HF	= (TEAC)				↓ (TBARS)		RCT	RW	M + F	H, Hypercholesterolaemic	?	21	10 weeks	red onion extract	250	Taiwan Tobacco and Liquor Corporation (TTL)
[90]	2017	Barden AE	↓					↑ F 2 -isoprostanes	RCT CO	RW / DRW	M	H	20–65	22	28	W	375	Institutional
[93]	2018	Di Renzo L	↑		↑ (SOD2)		=	↑ Antioxidant gene expression	RCT	RW/WW	M + F	H	18–65	54	acute	VDK, MeDM, HFM	30 g ET	The Ministry of Agriculture, Italy
[100]	2002	Choleva M			=(SOD2)	↓ 8-OH-dGuo ↓ 8-OH-Gua ↓ 8-NO2-Guo	= (TBARS)		RCT	RW	M + F	CAD	50–75	64	2 months	ET, W	200	Graduate Program of the Department of Nutrition and Dietetics, Harokopio University and the Hellenic Atherosclerosis Society.

↑, increased antioxidant effect; =, no effect; ↓, reduced antioxidant effect. 8-OH-dGuo, 8-hydroxy-2 deoxyguanosine; 8-OH-Gua, 8-hydroxyguanine; 8-NO2-Guo, 8-nitroguanosine; CD, conjugated dienes; CVD, cardiovascular disease; CO, crossover; ET, ethanol; F, females; FRAP, ferric reducing antioxidant power; GSH, glutathione; H, healthy; HFM, high-fat meal; LPD, low-phenolic diet; LOOH, lipid hydroperoxides; M, males; MDA, malondialdehyde; MeDM, Mediterranean meal; ORAC, oxygen radical absorbance capacity; PSRW, polyphenol-stripped red wine; RCT, randomised controlled trial; RW, red wine; SOD, superoxide dismutase; TAC, total antioxidant capacity; TAS, serum total antioxidant status; TBARS, thiobarbituric acid reactive substances; TEAC, Trolox equivalent antioxidant capacity; TRAP, total peroxyl radical-trapping antioxidant potential; VDK, vodka; W, water; WW, white wine. §, only DRW; °, only RW; &, one glass of RW displayed lower amounts of bioavailable flavonols compared with one glass of tea or 15 g of red onion, but similar in vivo and in vitro antioxidant activity; $ RW > DRW ≈ PSRW > W > ET; £ WW and RW exerted an antioxidant effect but the % inhibition of isoprostanes was RW > WW; # RW > DRW; ç, during a meal; ?, age was not provided.

**Table 2 nutrients-15-01921-t002:** Overview of studies that evaluated the cardiovascular effects of red wine consumption.

Ref.	Year	Author	IPC	Coronary	LV	RV	Carotid Plaque	Cerebral Blood Flow Velocity	Genes	Type of Study	Type of Wine	Subjects	Patients	Age Range	Total Number of Participants	Wine Consumption Duration (Days)	Control	Dosage (mL/Day)	Funding
[57]	2008	Marinaccio L	=							RCT	RW	M + F	Stable CVD	50–70	45	Acute	Gin, W	180 (18.9 g ET)	Not declared
[62]	2010	Kaul S		=						RCT	RW, WW	M + F	H	30–50	12/11/11/11	14	WW, W, VDK	355; 59 mL VDK	National Institutes of Health
[64]	2010	Kiviniemi TO		=						RCT CO	RW, DRW	M	H	20–25	22	Acute	DRW	120	Institutional
[65]	2011	Cameli M			↓	↑				RCT CO	RW	M + F	H	20–30	64	1	Fruit juice	337.6 ± 68.9 (0.5 g ET/kg)	Not declared
[76]	2014	Droste DW						=		RCT	RW	M + F	Carotid atherosclerosis	55–75	56/52	20 weeks	Med diet, no alcohol	F 100; M 200	Centre de Recherche Public-Santé
[94]	2018	Golan R					= §, ↑ ^			RCT (post hoc analysis)	RW and WW	M + F	TD2M	50–70	117/57	2 years	W	150: dry RW, (16.9 g ET), dry WW (15.8 g ET) or W	Institutional
[96]	2018	Roth I							↓	RCT CO	RW	M	High CVD risk	55–80	41	acute	Gin	AAW or gin (0.5 g ET/kg)	Fundación dela Investigación sobre Vinos y Nutricia

↑, improve; =, no effect; ↓, decline. AAW, Andalusian aged wine; CO, crossover; CVD, cardiovascular disease; DRW, de-alcoholised red wine; ET, ethanol; F, females; GSH, glutathione; H, healthy; IPC, ischemic pre-conditioning; LV, left ventricular function; M, males; RCT, randomised controlled trial; RO, red onion; RV, right ventricular function; RW, red wine; T2DM, type 2 diabetes mellitus; VDK, vodka; W, water; WW, white wine. §, no progression in carotid arterial total plaque volume; ^, in those with the greatest plaque burden, small regression of plaque burden; “coronary”, the coronary microcirculation or haemorrheology or coronary epicardial diameters or flow rate; “genes”, several genes related to the appearance and progression of atherosclerosis.

**Table 3 nutrients-15-01921-t003:** Effects of red wine consumption on the coagulation pathway.

Ref.	Year	Author	Thrombosis/Fibrinolysis System	Thrombotic Activation/Plasma Viscosity	Fibrinogen	PAF	PAI-1 §	vWF ^	Bleeding Time	D-Dimer	Type of Study	Type of Wine	Subjects	Patients	Age Range	Total Number of Participants	Wine Consumption Duration (Days)	Control	Dosage (mL)	Funding
[14]	2001	Ceriello A	↓	↓							RCT	RW	M + F	T2DM	50–60	20	7	W	300	Not declared
[17]	2002	Mansvelt EP		↓	=		↓				RCT	RW	M + F	H	25–56	13	28	WW	ET: 23 g for F, 32 g for M	Not declared
[20]	2003	Mezzano D				↑		=	=		RCT	RW	M	H	19–25	21	90	MD	240 (23.2 g ET)	University of Chile
[21]	2003	Pignatelli P						↓			RCT	RW, WW	M + F	H	35–50	24	14	WW	300	Not declared
[23]	2004	Kikura M		=		=					RCT CO	RW, WW	M + F	H	30–45	24	acute	CO	300–350	Not declared
[41]	2006	Jensen T		↓	↓						RCT CO	RW	M + F	H	35–70	92	21	W	150 (15 g ET)	Not declared
[58]	2008	Tousoulis D					=	↓			RCT	RW, WW	M + F	H	22–27	83	1	Beer (633 mL), whisky (79 mL) or W (250 mL)	264	University of Athens
[62]	2010	Kau S			=					=	RCT	RW, WW	M + F	H	30–50	12/11/11/11	14	WW, W, VDK	355; 59 VDK	National Institutes of Health, Bethesda, MD, USA
[72]	2013	Banach J					↑				RCT	RW, WW	M	H	20–30	12/11/11/12/11	5	WW, ET 12%, blackcurrant juice, W	300	Collegium Medicum of The Nicolaus Copernicus University
[87]	2016	Xanthopoulou MN				↑	=				RCT CO	RW vs. WW vs. ET	M	H	25–39	10	1	W	4 mL/kg BW	Institutional
[89]	2017	Argyrou C				↓					RCT	RW	M	H	25–39	10	1 (×4)	WW, ET, W	4 mL/kg BW	Graduate Program of the Department of Nutrition and Dietetics, Harokopio University.

↑, increased; =, no effect; ↓, decreased; BW, body weight; CO, crossover; ET, ethanol; F, females; H, healthy; M, males; MD, Mediterranean diet; PAF, platelet-activating factor; RW, red wine; T2DM, type 2 diabetes mellitus; VDK, vodka; W, water; WW, white wine; vWF, von Willebrand factor; §, a significant increase in the PAI-1:Ag concentration was observed in the RW-drinking group and a higher PAI-1 is a risk factor for thrombosis and atherosclerosis; ^, vWF promotes the adhesion of platelets to damaged blood vessel walls and then acts as a bridge between one platelet and another, acting as a glue and promoting clot formation.

**Table 4 nutrients-15-01921-t004:** Overview of studies that evaluated the effects of red wine consumption on endothelial function and arterial stiffness.

Ref.	Year	Author	Vasodilation	Arterial Stiffness	MSNA	FMD	FBF	Type of Study	Type of Wine	Subjects	Patients	Age Range	Total Number of Participants	Wine Consumption Duration (Days)	Control	Dosage (mL/Day)	Funding
[10]	2000	Agewall S	↑					RCT CO	RW	M + F	H	27–35	12	Acute	DRW	250	Swedish Medical Research Council, Swedish Medical Society
[25]	2004	Whelan AP				↑		RCT	RW, WW	M	CVD	30–70	14	Acute	CO	4	Southland Medical Foundation, Invercargill, New Zealand.
[27]	2005	Coimbra SR	↑			↑		RCT	RW	M + F	Hypercholesterolaemia	40–60	16	14 (×2)	Purple grape juice	250	FAPESP and Fundação Zerbini.
[29]	2005	Guarda E	=					RCT	RW	M + F	CVD	55–62	20	60	W	250	Not declared
[36]	2005	Zilkens RR				=		RCT CO	RW	M	H	20–65	24	28	DRW, beer	375	National Health and Medical Research Council of Australia
[40]	2006	Boban M	↑ §			=		RCT CO	RW	M	H	25–40	9	Acute	DRW, PSRW, 14% vol/vol ET, W	3 × BW	Not declared
[43]	2006	Naissides M		↑ *				RCT	RW	F	Hypercholesterolaemic postmenopausal	50–70	45	42	DRW, W	400 (40 g ET)	National Heart Foundation of Australia
[48]	2007	Karatzi K				↑		RCT CO	RW	M	Heavy smokers	22–24 and 66–75	12	Acute	DRW, SMOKING	250	Korea Research Foundation and USDA/ARS Western Human Nutrition Research Center at the University of California at Davis.
[49]	2007	Hijmering ML				↓		RCT	RW	M + F	H	25–45	20	Acute	Low-polyphenolic alcoholic fruit-flavoured drink	≃330	Not declared
[54]	2007	Spaak J			↑ ^	↑ ^		RCT	RW	M + F	H	24–47	13	14 (×3)	ET, W	155 (12% ET)	Heart and Stroke Foundation of Ontario and the Canadian Institutes of Health Research
[58]	2008	Tousoulis D					↑ ç	RCT	RW, WW	M + F	H	22–27	83	Acute	Beer (633 mL), whisky (79 mL) or W (250 mL)	264	University of Athens.
[61]	2010	Huang PH	↑					RCT	RW	M + F	H	30–40	80	21	W, beer, VDK	100	Institutional
[71]	2013	Barden AE	↓					RCT	RW	M	T2DM	20–65	25	Acute	DRW or W	375 (41 g ET)	Not declared
[72]	2013	Banach J	↓ £					RCT	RW, WW	M	H	20–30	12/11/11/12/11	5	WW, ET 12%, blackcurrant juice, W	300	Collegium Medicum of The Nicolaus Copernicus University
[90]	2017	Barden AE	↓ £					RCT CO	RW, DRW	M	H	20–65	22	28	W	375	Institutional

↑, improved; =, no effect; ↓, impaired. BW, body weight; CO, crossover; CVD, cardiovascular disease; DRW, de-alcoholised red wine; ET, ethanol; F, females; FBF, forearm blood flow; FMD, flow-mediated dilatation; H, healthy; M, males; MSNA, muscle sympathetic nerve activity, PSRW, polyphenol-stripped red wine; RCT, randomised controlled trial; RW, red wine; VDK, vodka; W, water; WW, white wine; ^, these acute effects are not modified by RW polyphenols; *, a pattern of improved vascular function and arterial stiffness was observed in the DRW group; §, the elimination of polyphenolic compounds from RW resulted in lack of the observed vasodilatation; Ç, increased maximum hyperaemic forearm blood flow 1 h after consumption, an effect persisting after 4 h; £, increased vasoconstrictor (endothelin-1).

**Table 5 nutrients-15-01921-t005:** Effects of red wine consumption on hypertension.

Ref.	Year	Author	SBP	DBP	SBP and DBP	Type of Study	Type of Wine	Subjects	Patients	Age Range	Total Number of Participants	Wine Consumption Duration (Days)	Control	Dosage (mL/Day)	Funding
[18]	2002	Foppa M			↓	RCT CO	RW	M + F	HPN, obese	35–65	13	Acute	W	250 (23 g ET)	Not declared
[31]	2005	Karatzi KN	↓			RCT CO	RW	M + F	CVD	40–60	15	Acute	DRW	250	Not declared
[36]	2005	Zilkens RR			↑	RCT CO	RW	M	H	20–65	24	29	DRW, beer	375	National Health and Medical Research Council of Australia
[44]	2006	Papamichael C	↓			RCT CO	RW	M + F	H, smokers	25–35	20	Acute	DRW	250	Not declared
[54]	2007	Spaak J			=	RCT	RW	M + F	H	24–47	13	Three occasions in 2 weeks	ET, W	155 (18.6 g ET); 310 for the second dose	Heart and Stroke Foundation of Ontario (T4938, T4050) and the Canadian Institutes of Health Research
[67]	2012	Chiva-Blanch G	↓	↓	↓	RCT CO	RW, DRW	M	high CVD risk	55–75	67	28	DRW, gin	30 g ET/d	Not declared
[69]	2012	Queipo-Ortuño MI			↓	RCT CO	RW	M	H	45–50	10	20	DRW, gin	272	Institutional
[71]	2013	Barden AE			↑	RCT	RW	M	T2DM	20–65	25	Acute	DRW or W	375 (41 g ET)	Not declared
[78]	2015	Fantin F	↑	=		RCT	RW	M + F	H	25–53	18	1	/	300 (12% ET)	Not declared
[79]	2015	Gepner Y			=	RCT	RW vs. W	M + F	T2DM	50–65	27/27	6 months	W	150 (16.9 g ET)	European Association for the Study of Diabetes
[82]	2015	Mori TA			↑ §	RCT CO	RW vs. DRW	F	H	25–49	24	28 × 3	High RW vs. low RW vs. DRW	200–300 (146–218 g ET/wk)	National Heart Foundation of Australia.
[86]	2016	Mori TA			= °	RCT CO	RW vs. DRW vs. W	M + F	T2DM	49–66	24	28 (per period)	RW vs. DRW vs. W	F: 230 (~24 g ET); M: 300 (~31 g ET)	Australian Health Management Group Medical Research Fund
[90]	2017	Barden AE			↑	RCT CO	RW/DRW	M	H	20–65	22	28	W	375	Institutional

↑, increased HPN; =, no effect; ↓, reduced HPN; BP, blood pressure; CO, crossover; CVD, cardiovascular disease; DBP, diastolic blood pressure; DRW, de-alcoholised red wine; ET, ethanol; F, females; H, healthy; HPN, hypertension; M, males; RCT, randomised controlled trial; RW, red wine; SBP, systolic blood pressure; T2DM, type 2 diabetes mellitus; W, water; WW, white wine; °, increased 24-h BP and HR but reduced waking BP; §, in the DRW group, BP values were similarly elevated.

**Table 6 nutrients-15-01921-t006:** Overview of studies that evaluated the effects of red wine consumption on immune function and inflammatory status.

Ref.	Author	IL-6	TNF-α	IFN-γ	hsCRP	Fibrinogen	Other Immune Parameters	Type of Study	Type of Wine	Subjects	Patients	Age Range	Total Number of Participants	Wine Consumption Duration (Days)	Control	Dosage (mL/Day)	Funding
[19]	Watzl B		=				= &	RCT	RW/ DRW	M	H	28–32	6	1	Red grape juice	500 mL of red wine (12% ET), a 12% ET dilution, DRW, and red grape juice	Federal Ministry of Consumer Protection, Food, and Agriculture, Germany.
[22]	Watzl B		=				= &	RCT	RW	M	H	25–35	24	14	ET 12%	500 (12% ET), 12% ET, DRW, and red grape juice,	Not declared
[26]	Williams MJ	↑					= £	RCT CO	RW and WW	M	CVD	48–70	13	Acute	CO	4 mL/kg	Not declared
[28]	Avellone G				↓		= )	RCT CO	RW	M + F	H	35–65	48	28 (×2)	Usual wine intake	250	Sicilian Agricultural Development Bureau, Palermo
[33]	Retterstol L					↓ °		RCT CO	RW	M + F	H	40–60	87	21	W	150 (15 g ET)	Sigurd K. Thoresen Foundation
[52]	Sacanella E						↓ ^	RCT CO	RW, WW	F	H	20–50	35	28 (×2)	W	200 (20 g ET)	Not declared
[55]	Vázquez-Agell M	↓			↓		↓ §	RCT CO	Cava (sparkling wine)	M	H	25–43	20	28	Gin	300 (30 g ET)	Spanish Ministries of Education and Science and Health
[58]	Tousoulis D	=	=		=	=		RCT	RW, WW	M + F	H	22–27	83	Acute	Beer (633 mL), whisky (79 mL) or W (250 mL)	264	University of Athens.
[66]	Chiva-Blanch G		=				↓ ç, = /	RCT CO	RW, DRW	M	High CVD risk	55–75	67	28	DRW, gin	30 g ET/d	Not declared
[72]	Banach J				=		↓ ç,↓ /	RCT	RW, WW	M	H	20–30	12/11/11/12/11	5	WW, ET 12%, blackcurrant juice, W	300	Collegium Medicum of The Nicolaus Copernicus University
[77]	Muñoz-González I	↓ *	↓ *	↓ *				RCT	RW	M + F	H	20–65	34/8	28	W	250 1758 mg of gallic acid equivalents/L and 12% ET w	Institutional
[90]	Barden AE						↓ <	RCT CO	RW/ DRW	M	H	20–65	22	28	W	375	Institutional
[92]	Barden AE						= >	RCT CO	RW/ DRW	M + F	T2DM	40–70	24	28 (×3)	No T2DM	W: 230 mL/day (~24 g ET/day)/M: 300 mL/day (~31 g ET/day)	Institutional
[95]	Wotherspoon A	=			=		= $	RCT CO	RW	M	H	21–70	77	28	Vodka	240	The Partridge Foundation
[97]	Fragopolou E		↓					RW	M + F	CAD	50–75	64	2 months	ET, W	200	Graduate Program of the Department of Nutrition and Dietetics, Harokopio University and the Hellenic Atherosclerosis Society.	

↑, increased; =, no effect; ↓, reduced; 20-HETE, 20-hydroxyeicosatetraenoic acid; CASP-1, caspase-1; CO, crossover; CVD, cardiovascular disease; DRW, de-alcoholised red wine; E-1 endothelin-1; ET, ethanol; hsCRP, high-sensibility C-reactive protein; H, healthy; ICAM-1, soluble intercellular adhesion molecule-1; IL, interleukin; MCP-1, monocyte chemoattractant protein-1; MMP-9, metalloproteinase-9; PAI-1, Ag plasminogen activator inhibitor antigen; RCT, randomised controlled trial; RW, red wine; t-PA Ag, tissue-type plasminogen activator antigen; SPMs, specialised pro-resolving mediators of inflammation; T2DM, type 2 diabetes mellitus; TGFb1, transforming growth factor-beta 1; TIMP-1, tissue inhibitor of metalloproteinase 1; VCAM-1, soluble vascular cell adhesion molecule-1; W, water; WW, white wine; *, only in patients with high cytokine levels; °, marginally reduced in healthy subjects; ^, vascular CAM-1 and E-selectin; §, VCAM-1, ICAM-1, E-selectin, P-selectin, MCP-1, and CD40L; ç, PAI-1:Ag; /, E-1, tPA:Ag; &, IL-2, IL-4; £ ICAM-1, VCAM-1; ), TGFb1; >, no effect on SPMs levels; <, increased 20- HETE and SPMs; $, IL-18, VCAM-1, CASP-1, MMP-9, TIMP-1, APO B, cystatin C, or ICAM-1.

**Table 7 nutrients-15-01921-t007:** Effects of red wine consumption on lipid profile and homocysteine levels.

Ref.	Year	Author	Cholesterol Efflux	RCTP	CM	Lipoprotein(a)	TG	TC	LDL	HDL-C	LDL/HDL	F2-Isoprostanes	Lipid Peroxidation	HCY	Type of Study	Type of Wine	Subjects	Patients	Age Range	Total Number of Participants	Wine Consumption Duration (Days)	Control	Dosage (mL/Day)	Funding
[11]	2000	Caccetta RA							=				=		RCT	RW	M	H	40–63	12	Acute	PSRW, DRW or W	5/kg BW	The National Heart Foundation of Australia and the Medical Research Foundation of Royal Perth Hospital
[12]	2000	Senault C	↓ *				=	↑ *	=	↑ / ↓ §					RCT CO	RW	M	H	18–35	56	14	DRW, ET	30 g of ET	ONIVINS and INSERM, Paris, France.
[13]	2001	Caccetta R					=	=	=	=		= / ↓ §			RCT	RW	M	H, smokers	25–71	18	14	WW, DRW	375	Australian Grape Wine Research and Development Corporation and the Medical Research Foundation of Royal Perth Hospital.
[16]	2001	van der Gaag MS		↓			=	=	=	↑				=	RCT CO	RW	M + F	H	44–59	11	21	Wine, beer, spirits, W	Four glasses (40 g ET)	Not declared
[24]	2004	Naissides M			= §		↑ / = §								RCT CO	RW	F	Dyslipidaemia postmenopausal	50–70	17	Acute (3 times in 2 weeks)	DRW, W	400	National Heart Foundation of Australia
[27]	2005	Coimbra SR					=	=	=	=					RCT	RW	M + F	Hypercholesterolaemia	40–60	16	14 (×2)	Purple grape juice	250	FAPESP and Fundação Zerbini.
[28]	2005	Avellone G				=	=	=	=	↑	↓				RCT CO	RW	M + F	H	35–65	48	4 + 4 weeks	Usual wine intake	250	Sicilian Agricultural Development Bureau
[30]	2005	Hansen AS								↓					RCT	RW	M + F	H	38–74	69	28	W + red grape extract, W + placebo	M: 300 mL/day, 38.3 g ET/day, F: 200 mL/day, 25.5 g ET/day	Not declared
[34]	2005	Tsang C					=	=	↓ *	↑ *					RCT	RW	M + F	H	23–50	20	14	W	375	Not declared
[35]	2005	Ziegler S							=						RCT	RW	M + F	H	22–32	60	Acute	2 RW, 1 WW	300	Jubiläumsfonds der Österreichischen National bank
[37]	2006	Banini AE					=	=	=	=					RCT	RW	M + F	T2DM, H	45–75	29	28	MJ, MW, or Dz-W	150	North Carolina Agricultural Research Service of the College of Agriculture and Life Sciences
[39]	2006	Blackhurst DM											=		RCT	RW	M + F	H	25–45	15	Acute	W	M: 230/F: 160	Winetech (Wine Industry Network of Expertise and Technology), Stellenbosch, South Africa
[47]	2007	Gorelik							↓ °						RCT CO	RW	Not declared	H	25–35	10	Acute	#	200	BARD, The United States-Israel Agricultural Research and Development Fund.
[56]	2008	Gibson A												↑	RCT CO	RW	M	H	21–70	78	14	Vodka	240 mL RW or 80 mL vodka (24 g ET)	The Pantridge Foundation; National Institutes of Health
[59]	2009	Estruch				↓	=	=	=	↑	↓				RCT CO	RW	M	H	30–50	40	28	Gin	30 g ET	Institutional
[63]	2011	Kechagias					=	=	↓	=					RCT CO	RW	M + F	H	25–45	52	3 months	W	W: 150 (16 g ET) M: 300 ML (33 g ET)	Institutional
[67]	2012	Chiva-Blanch				↓ / ↑ §	=	=	↓ ^	↑ ^	↓				RCT CO	RW, DRW	M	High CVD risk	55–75	67	28	DRW, gin	30 g ET	Institutional
[74]	2013	Droste				=	=	=	=	=	↓				RCT	RW	M + F	Carotid atherosclerosis	55–75	56/52	20 weeks	Med diet, no alcohol	F 100; M 200	Centre de Recherche Public-Santé
[86]	2016	Mori					=	↑	=	=					RCT CO	RW vs. DRW vs. W	M + F	T2DM	49–66	24	28	RW vs. DRW vs. water	W: 230 (~24 g ET); M: RW 300 (~31 g ET)	Australian Health Management Group Medical Research Fund
[88]	2016	Chiu					=	=	↓ ç	=					RCT	RW	M + F	H, Hypercholesterolaemic	?	21	10 weeks	RO extract	250	Taiwan Tobacco and Liquor Corporation (TTL)
[91]	2017	Taborsky					=	=	↓ [	=					RCT multi-centre	RW vs. WW	M + F	H	30–60	74/72	1 year	WW	200–300	Vino e Cuore, Ltd.
[98]	2022	Briansó-Llort L						↓ $	=	=					RCT CO	RW	M + F	H	30–50	26	28	RW	187	Instituto de Salud Carlos III
[100]	2002	Choleva M					=	=	=	=					RCT	RW	M + F	CAD	50–75	64	2 months	ET, W	200	Graduate Program of the Department of Nutrition and Dietetics, Harokopio University and the Hellenic Atherosclerosis Society.

↑, increased; =, no effect; ↓, reduced. BW, body weight; CM, chylomicrons; CO, crossover; DRW, de-alcoholised red wine; Dz-W, de-alcoholised muscadine grape wine; ET, ethanol; F, female; H, healthy; HCY, homocysteine; HDL-C, high-density lipoprotein; LDL, low-density lipoprotein; M, male; PSRW, polyphenol-stripped red wine; RCT, randomised controlled trial; RCTP, reverse cholesterol transport pathway; RO, red onion; RW, red wine; TG, triglycerides; W, water; WW, white wine; RTCP, the process by which cholesterol via HDL is directed from the peripheral tissues to the liver for subsequent excretion in the bile; lipoprotein (a), a type of lipoprotein that is responsible for transporting cholesterol in the blood and is considered a CVD risk factor that does not respond to standard LDL-lowering therapy; *, modest; §, DRW; °, inhibition of cytotoxic lipid peroxidation products (MDA); ^, only RW (not DRW); ç, results indicated that RO extract consumption rendered better cardioprotective effect than RW by altering cholesterol, improving antioxidation, and suppressing inflammatory marker levels, thereby, attenuating cardiovascular disease incidence; [, no difference compared with WW; #, A: 250 g turkey cutlets; W, B: soaked in RW after heating plus 200 mL of RW; C, soaked in RW prior to heating plus 200 mL of RW; $: only high resveratrol RW; ?, age was not provided.

**Table 8 nutrients-15-01921-t008:** Effects of red wine consumption on body weight, obesity, and adipocytokines.

Ref.	Year	Author	BMI	WC/Visceral Fat	Leptin	Adiponectin	Type of Study	Type of Wine	Subjects	Patients	Age Range	Total Number of Participants	Wine Consumption Duration (Days)	Control	Dosage (mL/Day)	Funding
[37]	2006	Banini AE	=	=			RCT	RW	M + F	T2DM, H (control)	45–75	29	28	MJ, MW, or Dz-W	150	College of Agriculture and Life Sciences
[38]	2006	Beulens JW		= *			RCT CO	RW	M + F	H, WC > 94cm	35–70	34	28	DRW	450 (40 g ETl)	Dutch Foundation for Alcohol Research
[46]	2007	Djurovic S			↑ §		RCT CO	RW	M + F	H	40–60	87	21	W	150 (15 g ET)	Not declared
[84]	2016	Golan R	=	=			RCT	RW	M + F	T2DM	40–75	27/21	2 years	W	150	Institutional
[95]	2020	Wotherspoon A			↑	=	RCT CO	RW	M	H	21–70	77	28	W	240	The Partridge Foundation
[98]	2022	Briansó-Llort L	=				RCT CO	RW ^	M + F	H	30–50	26	28	RW ^	187	Instituto de Salud Carlos III

↑, increased; = no, effect. CO, crossover; DRW, de-alcoholised red wine; Dz-W, de-alcoholised muscadine grape wine; F, females; H healthy; M, males; MJ, muscadine grape juice; MW, muscadine grape wine; RCT, randomised controlled trial; RW, red wine; T2DM, type 2 diabetes mellitus; W, water; WC, waist circumference; *, no fat accumulation in subcutaneous and abdominal fat depots in healthy subjects with a waist circumference above 94 cm; §, only females; ^, two different red wines with different resveratrol contents (low and high content).

**Table 9 nutrients-15-01921-t009:** Effects of red wine on glycaemic control in individuals with type 2 diabetes.

Ref.	Year	Author	Fasting Glucose	2 h Post-Meal Glucose	Fasting Insulin	Glycated Haemoglobin	Glucose/Insulin Ratio	HOMA-IR	ISI	Diabetic Nephropathy	SPMs	Type of Study	Type of Wine	Subjects	Patients	Age Range	Total Number of Participants	Wine Consumption Duration (Days)	Control	Dosage (mL/Day)	Funding/Conflict of Interest
[14]	2001	Ceriello A	↑		↑							RCT	RW	M + F	T2DM	50–60	20	7	W	300	Not declared
[27]	2005	Coimbra SR	=									RCT	RW	M + F	Hypercholesterolaemia	40–60	16	14 (×2)	Purple grape juice	250	FAPESP and Fundação Zerbini.
[37]	2006	Banini AE	=		↓ §	=	↑					RCT	RW	M + F	T2DM, H (control)	45–75	29	28	MJ, MW, or Dz-W	150	North Carolina Agricultural Research Service of the College of Agriculture and Life Sciences
[38]	2006	Beulens JW							=			RCT CO	RW	M + F	H, WC > 94 cm	35–70	34	28	DRW	450 (40 g ET)	Dutch Foundation for Alcohol Research.
[42]	2006	Marfella R	=		↓	=		↓				RCT	RW	M + F	T2DM, MI	30–40	131	1 year	W	118 (11 g ET)	Not declared
[53]	2007	Shai I	↓ °	=		=						Multi-centre RCT	RW or WW	M + F	T2DM	41–74	91	12 weeks	Non-alcoholic beer	150 (13 g ET)	Not declared
[60]	2009	Nakamura T				=				↓ urinary protein ↓ urinary L-FABP		RCT	RW, WW	M + F	T2DM with nephropathy	45–65	20	6 months	W	118	Not declared
[67]	2012	Chiva-Blanch G	=		↓T2DM (=H)			↓ T2DM (=H)				RCT CO	RW, DRW	M	High CVD risk	55–75	67	28	DRW, gin	30 g ET	Institutional
[80]	2015	Gepner Y	=					=				RCT	RW vs. WW vs. W	M + F	T2DM	50–70	224	2 years	RW vs. WW vs. W	150	European Association for the Study of Diabetes
[81]	2015	Moreno-Indias I	↓									RCT CO	RW and DRW	M	Obes, MeTs	45–50	10/10	30 + 30	H	RW or DRW: 272	Institutional
[92]	2018	Barden A									=	RCT CO	RW/DRW	M + F	T2DM	40–70	24	12 weeks (4 × 3)	No T2DM	W: 230 (~24 g ET/M: 300 (~31 g ET	Institutional
[98]	2022	Briansó-Llort L			=							RCT CO	RW	M + F	H	30–50	26	28	RW	187	Instituto de Salud Carlos III
[100]	2002	Choleva M	=		=			=				RCT	RW	M + F	CAD	50–75	64	2 months	ET, W	200	Graduate Program of the Department of Nutrition and Dietetics, Harokopio University and the Hellenic Atherosclerosis Society.

↑, increased; =, no effect; ↓, reduced. A1C, glycated haemoglobin; CO, crossover; CVD, cardiovascular disease; DRW, de-alcoholised red wine; Dz-W, de-alcoholised muscadine grape wine; ET, ethanol; F, female; H, healthy; HOMA-IR, homeostatic model assessment for insulin resistance; ISI, insulin sensitivity index; L-FABP, liver-type fatty acid binding protein (biomarker of kidney disease); M, male; MeTs, metabolic syndrome; MJ, muscadine grape juice; MW, muscadine grape wine; RCT, randomised controlled trial; RW, red wine; SPMs, specialised pro-resolving mediators of inflammation; T2DM, type 2 diabetes mellitus; W, water; WC, waist circumference; WW, white wine; §, decrease was significant only among the subjects with T2DM given Dz-W; °, especially in subjects with higher basal A1C levels.

**Table 10 nutrients-15-01921-t010:** Effects of red wine on the gut and its microbiota.

Ref.	Year	Author	Effects on the Composition of the Gut Microbiota	Steatosis	TMAO	Orocaecal Transit Time of Food	Meal-Induced Gallbladder Emptying	Gastric Emptying	Type of Study	Type of Wine	Subjects	Patients	Age Range	Total Number of Participants	Wine Consumption Duration (Days)	Control	Dosage (mL/Day)	Funding
[63]	2011	Kechagias S		= *					RCT CO	RW	M + F	H	25–45	52	3 months	W	W: 150 (16 g ET) M: 300 ML (33 g ET)	Institutional
[69]	2012	Queipo-Ortuño MI	↑ §						RCT CO	RW	M	H	45–50	10	20	DRW, gin	272	Institutional
[73]	2013	Clemente-Postigo M	↑						RCT CO	RW	M	H	45–50	10	20	DRW, gin	272	Institutional
[75]	2013	Kasicka-Jonderko A				=	↓	↓	RCT	RW	M + F	H	21–32	12	1	Beer, whiskey, W	200 (13.7 g ET)	Medical University of Silesia
[81]	2015	Moreno-Indias I	↑						RCT CO	RW and DRW	M	Obese, MeTs	45–50	10/10	30 + 30	H	RW or DRW: 272	Institutional
[83]	2016	Barroso E	=						RCT	RW	M + F	H	?	15/26	28	W	200	Institutional
[99]	2022	Haas EA	↑		=				RCT CO	RW	M	CAD	55–65	42	21	W	250 (5 d/wk)	São Paulo Research Foundation and others

↑, increase; =, no effect; ↓, decrease. CO, crossover; CVD, cardiovascular disease; Dz-W, de-alcoholised muscadine grape wine; ET, ethanol; F, female; H, healthy; M, male; MeTs, metabolic syndrome; RCT, randomised controlled trial; RW, red wine; TMAO, trimethylamine N-oxide; W, water; ? Sample age not provided; *, increased hepatic triglyceride content; §, only bifidobacteria.

## Data Availability

Data are available from the authors upon reasonable request.

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
