# Peer review of "Health Effects of Red Wine Consumption: A Narrative Review of an Issue That Still Deserves Debate"

_nutrients, 2023, doi:10.3390/nu15081921_

Round 1

Reviewer 1 Report

Very nice, well written review discussing the health effects of red wine consumption.

My only comment is that it would be nice for the complicity of the article to include a chapter regarding the effects of red wine on atrial fibrillation and other cardiac arrhythmias (atrial ectopy/atrial tachycardia and ventricular ectopy/tachycardia).

In addition, a comparison of the effects or RW, DRW and non-wine spirits on cardiac arrhythmias would be useful.

Author Response

Reviewer 1 

Very nice, well written review discussing the health effects of red wine consumption.

My only comment is that it would be nice for the complicity of the article to include a chapter regarding the effects of red wine on atrial fibrillation and other cardiac arrhythmias (atrial ectopy/atrial tachycardia and ventricular ectopy/tachycardia).

In addition, a comparison of the effects of RW, DRW and non-wine spirits on cardiac arrhythmias would be useful.

We thank the Reviewer for the useful and appreciating comments. 

We re-read the papers included in the review. Unfortunately, we did not find any RCTs evaluating the risk of arrhythmia associated with RW consumption.
However, we have added a paragraph concerning these important aspects to the discussion as requested.

Reviewer 2 Report

The review article collects data on the positive and negative health effects of red wine consumption. The article, of great scientific relevance, and not only, at this moment, is well written and clear and explores various aspects: antioxidant status, cardiovascular function, coagulation pathway and platelet function, endothelial function and arterial stiffness, hypertension, immune function and inflammation status, lipid profile and homocysteine levels, body composition, type 2 diabetes and glucose metabolism, and gut microbiota and gastrointestinal tract.

The search criteria and selection or exclusion of articles are well explained, as well as the set of results for each category. unfortunately the tables are difficult to read and should be made clearer. the conclusions and the discussion are consistent with what is reported in the results.

Author Response

The review article collects data on the positive and negative health effects of red wine consumption. The article, of great scientific relevance, and not only, at this moment, is well written and clear and explores various aspects: antioxidant status, cardiovascular function, coagulation pathway and platelet function, endothelial function and arterial stiffness, hypertension, immune function and inflammation status, lipid profile and homocysteine levels, body composition, type 2 diabetes and glucose metabolism, and gut microbiota and gastrointestinal tract.

The search criteria and selection or exclusion of articles are well explained, as well as the set of results for each category. unfortunately the tables are difficult to read and should be made clearer. the conclusions and the discussion are consistent with what is reported in the results.

We deeply appreciate the Reviewer's positive feedback.

Indeed, the tables are difficult to read, there is a lot of data. We moved the tables to another file and put them all horizontally. We hope that they are now more readable. 

The manuscript has been proofread by a professional proofreading company.

Reviewer 3 Report

The present manuscript is a well conducted and reported review of randomized clinical trials assessing the effect of red wine consumption on health. It is an exhaustive revision of articles published since 2000 summarizing the existing knowledge on this important topic.

There are no major concerns related to the review, only a recommendation to include tables in horizontal format, since in their present setting is almost impossible to properly see the information contained in them.

In line 147 it is stated that homovanillic acid is a typical biomarker of dopamine turnover, which is correct, but authors should also acknowledge that this may also be a phenolic metabolite, thus increased blood concentrations of homovanillic acid might also derived from metabolism of RW phenolic compounds.

Authors should carefully revise the abbreviations making sure to define (spell out) each abbreviature the first time they appear in the text and always in the tables.

Other minor issue is the need to homogenize how volume units are given (mL is the proper nomenclature) and abbreviations (systolic and diastolic blood pressure are given as SBP or DBP but also as systolic BP and diastolic BP).

Figure 2 is not included in the manuscript (unless it corresponds to the “Summary Points list”?

Author Response

The present manuscript is a well conducted and reported review of randomized clinical trials assessing the effect of red wine consumption on health. It is an exhaustive revision of articles published since 2000 summarizing the existing knowledge on this important topic. There are no major concerns related to the review, only a recommendation to include tables in horizontal format, since in their present setting is almost impossible to properly see the information contained in them.

We thank the Reviewer for the useful and appreciating comments. 

Indeed, the tables are difficult to read, there is a lot of data. We moved the tables to another file and put them all horizontally. We hope that they are now more readable. 

In line 147 it is stated that homovanillic acid is a typical biomarker of dopamine turnover, which is correct, but authors should also acknowledge that this may also be a phenolic metabolite, thus increased blood concentrations of homovanillic acid might also derived from metabolism of RW phenolic compounds.

We thank the reviewer for this important clarification. We have added the information on homovanillic acid in the manuscript. 

Authors should carefully revise the abbreviations making sure to define (spell out) each abbreviature the first time they appear in the text and always in the tables. Other minor issue is the need to homogenize how volume units are given (mL is the proper nomenclature) and abbreviations (systolic and diastolic blood pressure are given as SBP or DBP but also as systolic BP and diastolic BP).

We fixed the abbreviations and entered all volume units as mL. We have added at the end of the paper before the references a list of abbreviations used in the text. 

Figure 2 is not included in the manuscript (unless it corresponds to the “Summary Points list”?

Figure 2 was indeed removed from the text and we decided to use it as a graphical abstract. We have removed the reference to figure 2 from the manuscript.